# Mouse Models for Immune Checkpoint Blockade Therapeutic Research in Oral Cancer

**DOI:** 10.3390/ijms23169195

**Published:** 2022-08-16

**Authors:** Wei-Chiao Chiu, Da-Liang Ou, Ching-Ting Tan

**Affiliations:** 1Department of Medical Research, Fu-Jen Catholic University Hospital, Fu-Jen Catholic University, New Taipei City 24352, Taiwan; 2Department of Otolaryngology, National Taiwan University Hospital, Taipei City 100225, Taiwan; 3Graduate Institute of Oncology, College of Medicine, National Taiwan University, Taipei City 10051, Taiwan; 4YongLin Institute of Health, National Taiwan University, Taipei City 10672, Taiwan; 5Stem Cell Core Laboratory, Center of Genomic Medicine, National Taiwan University, Taipei City 10051, Taiwan; 6Department of Otolaryngology, College of Medicine, National Taiwan University, Taipei City 100233, Taiwan; 7Department of Otolaryngology, National Taiwan University Hospital Hsin-Chu Branch, Hsinchu 302058, Taiwan

**Keywords:** oral squamous cell carcinoma, immune checkpoint inhibitors, syngeneic tumor models, chemical carcinogen induction, genetically engineered mouse, humanized mouse, tumor microenvironment

## Abstract

The most prevalent oral cancer globally is oral squamous cell carcinoma (OSCC). The invasion of adjacent bones and the metastasis to regional lymph nodes often lead to poor prognoses and shortened survival times in patients with OSCC. Encouraging immunotherapeutic responses have been seen with immune checkpoint inhibitors (ICIs); however, these positive responses to monotherapy have been limited to a small subset of patients. Therefore, it is urgent that further investigations into optimizing immunotherapies are conducted. Areas of research include identifying novel immune checkpoints and targets and tailoring treatment programs to meet the needs of individual patients. Furthermore, the advancement of combination therapies against OSCC is also critical. Thus, additional studies are needed to ensure clinical trials are successful. Mice models are advantageous in immunotherapy research with several advantages, such as relatively low costs and high tumor growth success rate. This review paper divided methods for establishing OSCC mouse models into four categories: syngeneic tumor models, chemical carcinogen induction, genetically engineered mouse, and humanized mouse. Each method has advantages and disadvantages that influence its application in OSCC research. This review comprehensively surveys the literature and summarizes the current mouse models used in immunotherapy, their advantages and disadvantages, and details relating to the cell lines for oral cancer growth. This review aims to present evidence and considerations for choosing a suitable model establishment method to investigate the early diagnosis, clinical treatment, and related pathogenesis of OSCC.

## 1. Introduction

Oral cancer is one of the most common malignant neoplasms in humans, endangering human health, of which more than 90% are oral squamous cell carcinomas (OSCC).

OSCC is a subtype of head and neck squamous cell carcinoma (HNSCC) with an estimated incidence of more than 370,000 new cases and 170,000 deaths annually [1,2,3,4]. Globally, the main risks of OSCC are tobacco smoking, alcohol drinking, and betel nut chewing, followed by infection with high-risk human papillomavirus (HPV) [5,6,7,8]. OSCC usually occurs in elderly patients; however, the incidence has increased in young people and is mainly due to HPV-associated oropharyngeal squamous cell carcinoma [9]. Over half of OSCC patients are diagnosed at the T3 or T4 stage of disease progression, during which cancer invades local bones of the maxilla or mandible or metastasizes to regional lymph nodes [10]. In addition to lymph node metastasis, local infiltration of submucosa and bone is a common histological feature of OSCC [11,12]. Different types of bone invasion, including erosive, infiltrative, and mixed patterns, have been found in OSCC, which have different histological features and 3-year disease-free incidence. Compared with the erosive bone invasion pattern of OSCC, infiltrative bone invasive OSCC has a lower rate of 3-year disease-free status [13].

So far, despite the advanced technology in surgery, chemotherapy, and radiotherapy, the survival rate has hardly improved in the past two decades. The tumor microenvironment (TME) contains many different normal cells that have an essential role in tumor development and progression. The stromal fibroblasts, extracellular matrix, blood vessels, lymphatic vessels, infiltrating immune cells, growth factors, and cytokines secreted by TME cells all have positive and negative effects on tumor development [14,15,16,17,18].

Immunotherapy was developed through advances in knowledge of the interaction between the immune system and tumors and has improved treatment prospects in cancer patients. The methods of immunotherapy assist the immune components in the TME to resist the ability of the tumor to escape immune surveillance, by which the innate immune cells eliminate cancer cells or enhance the anti-tumor immune response [19]. The immune checkpoint blockade (ICB) approach, one of the immunotherapies, aims to drive the immune system to generate an effective anti-tumor response [20,21]. Immune checkpoint inhibitors (ICIs) are a new kind of anti-tumor immunotherapeutic agent that can inhibit many immune checkpoints, especially on cytotoxic T cells [22,23]. Identifying novel immune checkpoints and targets and tailoring treatment to individual patients is one focus area in immunotherapy. The immunotherapeutic effects of immune checkpoint inhibitors have been encouraging; however, only a limited subset of patients respond to monotherapy. Therefore, it is urgent to carry out further research, develop new combination therapies, develop more immunotherapeutic drugs, and improve the success rate of clinical trials.

To evaluate the effectiveness of immunotherapy, animal models need to be established in which human tumors and their microenvironment are genetically, physiologically, and anatomically modeled in order to faithfully reflect the formation and development of human tumors. Compared with other animal models, mice have lower costs, shorter reproductive cycles, higher tumor growth rates, and easy genetic modification. Furthermore, established inbred mice allow for tumor transplantation among the same strain of mice or cell lines from the same strain of mice. These advantages make the mouse model a good tool for evaluating the effectiveness of cancer immunotherapy. However, the ability to transfer encouraging immunotherapy results from preclinical trials to the clinic is now a challenge because, after promising results in mouse models, high failure rates have been observed in human clinical trials [24,25].

Oral carcinogenesis is a complex process composed of oral carcinogens (i.e., alcohol, tobacco, betel nut) and/or human HPV caused by a variety of genetic and epigenetic changes. The tumor microenvironments in mouse models that mimic the human cancer growth genetically, physiologically, and anatomically are important for the research of OSCC immunotherapy [24,26,27]. While a single model cannot recapitulate all aspects of OSCC, the data gathered from animal models are vital for advancing OSCC diagnosis and treatment [28,29].

This review presents an overview of current oral cancer mouse models and discusses the interplay between oral cancer biology and the immune system and regulators. In addition, we also present phase III trials that have been evaluated using one or a combination of immune checkpoint inhibitors in HNSCC (Table 1). Moreover, we review the different immunotherapeutic mouse models in use (Figure 1), (their advantages and disadvantages (Table 2)) and we detail cell strains used for inducing oral cancer growth (Table 3).

## 2. Syngeneic Mouse Models

Syngeneic OSCC mouse models are produced using allografts of immortalized mouse tumor cell lines. The models can efficiently prevent tissue rejection or graft-versus-host disease (GvHD), which results from transplanting tumor cells into mice. In addition, these models have the inherent advantage of fast establishment, high stability, and high consistency of transplanted tumors. Therefore, they are widely used in immuno-oncology studies and show great potential in developing novel OSCC treatments, especially immunotherapy [70]. Tumor cells can be injected orthotopically or ectopically. The orthotopic model supplies a more exact tumor microenvironment, while the subcutaneous model facilitates tumor monitoring and handling. For oral cancer induction, orthotopic models are performed by injecting tumor cells into the oral cavity, while ectopic models most commonly receive subcutaneous injections in the flanks [24].

The principle of syngeneic models is similar to the cell-derived xenograft (CDX) model. Several mouse cell lines can be used to develop syngeneic OSCC mouse models, including mouse OSCC Sq-1979 cells [36], mouse squamous cell carcinoma SCC7 [37,38], mouse oral cancer (MOC) cell lines [40], MOC1 [41], and MOC2 [42]. Moroishi et al. [39] have demonstrated that transplanting SCC7 cells (1 × 10^5^) into both hind flanks of C3H/HeOu mice resulted in aggressive tumor growth, whereas LATS1/2 dKO SCC7 cells did not result in tumors. Similarly, Dong et al. [38] injected SCC7 cells (1 × 10^6^) into the abdomen of C3H/HeJ mice to assess the effectiveness of a tumor-derived autophagosome vaccine (DRibble). Nagaya et al. [40] studied the effects of near-infrared photoimmunotherapy using syngeneic models developed through the subcutaneous injection of C57BL/6 mice with poorly immunogenic MOC2 mKate2 cells (1.5 × 10^5^), moderately immunogenic MOC2-luc cells (1.5 × 10^5^), and immunogenic MOC1 cells (2.0 × 10^6^). Similarly, Adachi et al. [36] injected Sq-1979 cells (1 × 10^7^) into the posterior neck area of C3H/HeN mice to determine the genetic changes that occur throughout OSCC development. These studies demonstrate that different mouse OSCC cell lines could be successfully used to produce stable syngeneic OSCC models.

In addition, several researchers use 4-nitroquinoline-1-oxide (4NQO) to induce OSCC in a mouse and then inject the OSCC obtained from the mouse into another mouse of the same species to establish a syngeneic model. This is similar to the construction of patient-derived xenograft (PDX) models. For example, Chen et al. used 4NQO (100 μg/mL in drinking water) to induce C57BL/6 mice over 16 weeks; the mice were sacrificed at week 28 to generate the mouse tongue OSCC cell lines MTCQ1 and MTCQ2. Afterward, the MTCQ1 or MTCQ2 cells were injected into the flank or tongue of new mice to establish ectopic or orthotopic mouse models, respectively [43]. Compared with human SAS tongue SCC cell lines, MTCQ cells have lower proliferation ability but far higher abilities of migration/invasion. Such capabilities are demonstrated through the identification of extensive cervical lymph node metastasis and lung metastasis resulting from an MTCQ1 cell subclone. Several therapeutic approaches have been tested using this model, including anti-PD-L1 immunotherapy, cisplatin therapy, and miRNAs (especially miR-134). In addition, Chen et al. [44] established a syngeneic model using 4NQO-induced OSCC transgenic mice. In this model, K14-EGFP-miR-211 transgenic mice were induced using 4NQO (100 μg/mL in drinking water) over 16 weeks and then sacrificed. Tissue isolated from OSCC lesions on the dorsal surface of the tongue was then used to produce cell lines MOC-L1 to -L4. These were subsequently used to create orthotopic xenografts and real-time in vivo tumor imaging by injecting cells (5 × 10^6^) into the central tongue of C57BL/6 mice. These cells were also used to measure the efficacy of cisplatin therapy and study distant metastasis. Chen et al. [45] established the NHRI-HN1 and NHRI-HN2 cell lines from 4-NQO/arecoline-induced murine tongue tumors and further selected for cell stemness by in vitro sphere culture to evaluate potential immunotherapy for OSCC in East and Southeast Asia. NHRI-HN1 or NHRI-HN2 cells (5 × 10^5^) in 50 μL in sterile phosphate-buffered saline (PBS) were injected into the buccal mucosa of mice that were sacrificed 40 days after injection.

Syngeneic tumor models have also been applied to the investigation of the anti-tumor activity of ICIs, including anti-programmed death (PD)-1/anti-PD-ligand 1 (L1) antibodies [71,72] and anti-cytotoxic T lymphocyte-associated antigen 4 (CTLA-4) [46]. The time taken to produce syngeneic tumor models is short, as tumor growth happens within a few weeks [25,73]. However, such rapid tumor growth can prevent the assessment of immunotherapeutics, as the treatment effect is often progressive and estimated by improving survival [74]. This makes syngeneic models unsuitable for assessing immunotherapy drugs at the early stages of tumor development [75]. The syngeneic OSCC mouse model is a viable tool for immuno-oncology. Still, the main problem is that the model only represents mouse oral cancer and forms mouse tumors with mouse targets. Mice and humans differ in compositions and mechanisms, and some targets in humans are absent or unresponsive in mice.

## 3. Chemotoxic Agent Mouse Models

Chemical carcinogen-induced (CI) tumor models are easier to establish experimentally and the cancers induced by carcinogen treatment are more like sporadic human cancers [73,76]. The mechanism of action can be divided into genotoxic (GTX) and non-genotoxic (NGTX). In the part of GTX, carcinogen directly acts on DNA or chromosomes and injures them [77], while the mechanism of NGTX action is first to trigger many other cellular effects, thus leading to changes in intracellular signaling pathways [78,79].

GTX carcinogens have electrophilic properties and can covalently bind to DNA to form DNA adducts and thus cause mutations [80,81]. GTX carcinogens can be said to be the initiator of cancer. For example, benzo(a)pyrene (BaP), a tobacco-related carcinogen, can form bulky BaP-DNA adducts and basic sites or, by reactive oxygen species (ROS) and metabolites generated during metabolism, causes DNA damage [82,83]. In contrast, NGTX carcinogens do not directly bind to DNA but affect the cell cycle, cause chronic inflammation [84], activate steroid hormone receptors (SHRs) to activate silent genetic pathways and generate reactive oxygen species (ROS), or cause immunosuppression [85]. In general, NGTX carcinogens have different actions, including mitogenic promotion (phenobarbital) [86], affecting the receptor-interacting protein-mediated pathway (2,3,7,8-tetrachlorodibenzo-p-dioxin) [84,87,88], cytotoxicity (asbestos) [89], or interfering with gap junction intercellular communication (chloroform) [90]. The epigenetic mechanisms by these NGTX carcinogens cause genetic instability, which in the corresponding microenvironment leads to tumorigenesis [91].

CI models are usually established for a long time, but they have great genomic complexity, which accurately reflects the real situation of human tumorigenesis. Furthermore, with the larger mutational burden, neoantigens with different degrees of immunogenicity may be generated, thus affecting the immunogenicity of the tumor [92]. The generation of unknown neoantigens makes it challenging to determine and affect immune responses using the CI model. As such, these models are rarely utilized to assess the effectiveness of ICIs. However, the mouse model of oral cancer induced by 4NQO has been established and used to evaluate the efficacy of the anti-PD-1 antibody [93,94].

Chemical risk factors for OSCC include arecoline, tobacco, and alcohol [95,96]. There are more than 60 cancer-causing agents in cigarette smoke, including polycyclic aromatic hydrocarbons (PAHs, such as benzo(a)pyrene), tobacco-specific nitrosamines (such as N’-nitrosamines (NNN) and 4-(methyl nitrosamine)-1-(3-pyridyl)-1-butanone (NNK)), and aromatic amines such as 4-aminobiphenyl [97]. These chemicals have been reported to induce DNA adducts (DNA covalently bound to carcinogens, related substances, or their metabolites) and to be associated with cancer susceptibility [97,98]. The researchers have induced oral carcinogenesis in mice using various chemical carcinogens, such as 4NQO [99], dibenzo(a,l)pyrene (DB(a,l)P) [58], benzo(a)pyrene (B(a)P) [59], NNN [60], and combinations of arecoline or ethanol with 4NQO [56,57]. Exposure of the mouse oral cavity to these chemical carcinogens can be used to produce primary OSCC mouse models.

### 3.1. 4NQO-Induced OSCC Mouse Model

Immunocompetent C57BL/6, BALB/c, CF-1, and CBA mice can be used to generate 4NQO-induced OSCC mouse models. Most studies applying this approach have used 6–8-week-old adult mice, but Vincent-Chong et al. noticed that 92% of aged mice (65–70 weeks of age) developed severe dysplasia/invasive squamous cell carcinoma, compared with 69% of young mice (7–12 weeks of age) [100]. The exposure of C57BL/6 and CBA mice to 4NQO in drinking water produced multiple precancerous lesions and carcinogenesis on the esophagus and tongue. CBA mice were more sensitive to 4NQO induction than the C57BL/6 strain [101,102].

The administration of 4NQO can be either topical or in drinking water. Both methods successfully establish OSCC mouse models [100,103,104,105,106,107]. Initially, researchers used a topical daub of 4NQO to induce OSCC, which was the same method as using DMBA to induce SCC in hamster pouches [103,106]. Schoop et al. applied 4NQO (5 mg/mL) dissolved in propylene glycol to the tongues of male CBA mice three times a week over 16 weeks [103]. Such treatment produced continued development of hyperplasia; mild, moderate, and severe dysplasia; and squamous cell carcinoma between weeks 24 to 40. This method ensured that 4NQO was localized to the oral cavity and reduced digestive tract exposure. In contrast, using 4NQO-containing drinking water to model OSCC is considered a more natural mode of administration, faster to model, and less painful to mice than topical application [100,104,105,107].

The amount and timing of 4NQO used in the drinking water method varied between studies. The concentrations of 4NQO most commonly used to induce OSCC in drinking water are 50 and 100 μg/mL [47,48]. A higher 4NQO concentration results in faster OSCC establishment. Mice are typically exposed to 4NQO-containing drinking water for 16 weeks (treatment period), followed by normal drinking water for 6–16 weeks (developmental period) [100,104,105,107]. At present, no standardized method exists for producing 4NQO-induced OSCC mouse models, and researchers can adjust these parameters according to the needs of the study. Tang et al. gave CBA mice different concentrations of 4NQO in drinking water: 100 μg/mL (16 weeks), 100 μg/mL (8 weeks), 50 μg/mL (8 weeks), and 20 μg/mL (8 weeks). All mice in the 100 μg/mL, and 50 μg/mL groups formed oral lesions, and the malignancy of oral lesions were 21.0 ± 0.9%, 15.1 ± 0.8%, 10.9 ± 0.8%, and 11.1 ± 0.8%, respectively [101].

The 4NQO-induced OSCC mouse model is widely used in research, given its similarities to human disease at the molecular level and regarding host immune activity, pathological changes, and pathogenesis. This model is particularly beneficial in the development of biomarkers for early diagnosis and translation of the epithelium of the material. Furthermore, the pathogenesis of OSCC can be studied using 4NQO as it can induce primary OSCC and mimics tobacco-related gene mutations [108]. The advantages of primary OSCC is that it closely mimics the tumor microenvironment in humans [100,109] and interacts with cytokines [47], mesenchymal stem cells [110], natural killer (NK) cells [111,112], microbiomes [113], and angiogenesis [114]. As a result, such lesions can be used to investigate the early identification and treatment of precancerous OSCC lesions [94,115,116]. In addition, given the similarities in host immune activity, this model allows for tumor immunology during tumorigenesis and development and related topics such as immunosuppression [49,117,118,119] and immunotherapy to be investigated [99,120,121]. However, a limitation of this model is the low probability of metastasis and bone invasion, limiting its use for investigating malignant tumor invasion and metastasis. In conclusion, given the model’s similarities to human OSCC it is best suited to investigating diagnostic and prognostic markers of OSCC.

### 3.2. Tobacco-Related Chemical Carcinogens-Induced OSCC

Chemical carcinogens associated with tobacco smoke include B(a)P, DB(a,l)P, and NNN. B(a)P is a member of the family of PAHs found in tobacco smoke, charcoal-grilled foods, contaminated water, engine exhaust gas, and soil. DB(a,l)P is a potent carcinogen produced in cigarette smoke. NNN is a nitrosamine generated during the curing and subsequent processing of tobacco. The researchers used these carcinogens to induce tumorous formation in mice and found that cancer incidence in other parts of the mouse body, including the liver, lungs, forestomach, esophagus, and larynx, was higher than in the tongue, thus limiting their application on OSCC models. (Table 3) [58,59,60,122,123,124,125].

### 3.3. 4NQO Combined with Other Chemical Carcinogens

Researchers have used 4NQO combined with other carcinogens, such as arecoline or alcohol to induce oral cancer. Arecoline, a betel nut-derived alkaloid, plays a crucial role in oral cancer progression. Alcohol is another risk factor for oral cancer. 4NQO, arecoline, or alcohol alone have little effect on inducing oral cancer in mice, but when arecoline or alcohol is combined with 4NQO, the incidence of tongue tumors in mice can even reach 100% (Table 3) [56,57,126,127,128,129].

In general, carcinogens cannot induce specific genes during oral carcinogenesis nor can they induce tumors at specific sites, so the CI models can be assisted by xenograft models or transgenic mouse models to meet individual research needs.

## 4. Genetically Engineered Mouse Models

Genetically engineered mouse models (GEMMs) include “loss-of-function” and “gain-of-function” categories. “Loss of function” uses gene knockout or knockdown technology to deplete or silence target genes such as oncogenes, tumor-suppressor genes, or metabolic genes. “Gain-of-function” uses the knock-in technique to overexpress oncogenes for the function study of oncogenes in vivo. According to the particularity, GEMM is divided into normal GEMM or conditional GEMM. Normal GEMM alters target genes throughout the entire organism. However, this method is inconsistent with tumor formation, given that tumorigenesis is triggered by a single cell in which multiple mutated gene loci inhibit apoptosis and promote proliferation [129,130]. Conditional GEMM is spatiotemporally specific, and the use of this technique can specifically alter target genes in different tissues (space) or at different periods (time). The resulting pathological changes were similar to those of primary human OSCC and mouse OSCC induced by 4NQO. Taking L2D1^+^/p53^+/−^ and L2D1^+^/p53^−/−^ mice as examples, at 5–6 months old, the histological examination demonstrated hyperplasia, hyperkeratosis, severe epithelial dysplasia, and cancer [61]. GEMM features an immunocompetent environment allowing tumors to grow, making this model suitable for immuno-oncology research [70]. C57BL/6 was the first mouse strain to undergo whole-genome sequencing and is considered the “standard” inbred strain that provides the genetic background for many targeted genes. C57BL/6 is often used in genetic experiments as a transgenic mouse model to recreate human genetic defects. In addition to C57BL/6 mice, Balb/c mice can also be used as the genetic background for GEMMs [41,70].

### 4.1. LSL-Kras^G12D^ Mice

Two oral cancer GEMMs are known to be associated with K-*ras* mutations. Researchers use the keratin 14 (K14) or keratin 5 (K5) promoters, which tend to express in the oral cavity, to drive overexpression of the K-*ras*^G12D^ oncogene in mouse oral epithelial cells [62,131]. Caulin et al. demonstrated that the K-*ras*^G12D^ oncogene driven by the K14 or K5 promoter is distributed in a modified recombinase, Cre, fused to a deletion mutant human progesterone receptor, which fails to bind its ligand progesterone but can be activated by progesterone antagonists, such as RU486. The treatment of RU486 caused K-*ras*^G12D^ overexpression in mouse oral epithelial cells and developed oral squamous papilloma [62]. In an additional model, the K5 promoter-driven expression of the K-*ras*^G12D^ oncogene is controlled by *tet*-responsive elements. The administration of doxycycline to mice can also be used to induce overexpression of K-*ras*^G12D^ [131]. Such overexpression results in dysplasia and SCC affecting the tongue, oral mucosa, esophagus, forestomach, or skin.

### 4.2. L2D1^+^/p53^+^^/^^−^ and L2D1^+^/p53^−/−^ Mice

Opitz et al. designed the first GEMM specifically for the development of HNSCC. They used the ED-L2 promoter (L2) from the Epstein–Barr virus to specifically modify genes in squamous epithelial cells in the mouth and esophagus. These modifications produced L2-cyclin D1 (L2D1^+^) mice, which contained L2 fused to human cyclin D1 cDNA. These mice were further crossed with p53^+/−^ and p53^-/−^ mice to produce composite mice predisposed to aggressive oral–esophageal SCC. In addition, cell lines generated from the oral epithelial cells of these composite mice could be used to induce tumors in athymic nu/nu mice [61,132].

### 4.3. Tgfbr1/Pten 2cKO (Tgfbr1^flox/flox^; Pten^flox/flox^; K14-CreER^tam^) Mice

*Tgfbr1*/*Pten* double conditional knockout (2cKO) mice can be induced in 4 weeks with tamoxifen (tam) to produce hyperplastic changes affecting the oral epithelium. Furthermore, after ten weeks, visible tumors develop that are well-differentiated SCC, mimicking HNSCC. *PTEN* is a negative PI3K/Akt pathway regulator and tumor suppressor gene. The interaction between PTEN and the PI3K/Akt pathway is crucial in HNSCC carcinogenesis. In contrast, TGF-β signaling is understood to be both tumor-promoting and tumor-suppressing and affects the carcinogenesis of various cancers. In *Tgfbr1*/*Pten* 2cKO mice, the loss of PTEN produces epithelium hyperproliferation and the loss of Tgfbr1 causes immune suppression and increases inflammation-enhanced angiogenesis. Thus, this model suggests that HNSCC carcinogenesis is affected by oncogenic epithelial changes and the tumor microenvironment [63].

### 4.4. p53^R172H^; K5.CrePR1 and p53^flox/flox^; K5 CrePR1 Mice

In the *p53^R172H^*; *K5.CrePR1* and *p53^flox^*^/*flox*^; *K5 CrePR1* models, cutaneous SCC occurred more frequently than oral SCC and there were metastases. In contrast, the incidence of OSCC in *p53^flox^*^/*flox*^; *K5 CrePR1* mice was 25% and only 16% in *p53^R172H^*; *K5.CrePR1* mice developed OSCC. In both models, deletion of the *Cdkn2a* gene enhanced skin tumor formation and promoted metastasis. However, the role of this gene in oral tumors remains to be explored. The gain and the loss of the p53 function were observed to be associated with carcinogenesis in HNSCC in both models [64].

In conclusion, the tumor microenvironment of GEMMs in mice differs significantly from humans due to the presence of transgenes in the stromal cells. Transgene expression can be localized through oral mucosa-specific promoters, such as the K14 or K5, to increase the specificity of the model. However, this does not entirely prevent leaky expression in other tissues. In addition, the natural carcinogenesis of oral cancer is never predominated by a single gene. Therefore, using specific genes (such as *Akt* or *K-ras*) to produce tumors in transgenic mice may differ to human oncogenic processes. This observation is aptly demonstrated by the low frequency of *K-ras* mutations found in human head and neck cancers [133,134].

## 5. Humanized Mouse Models

The above syngeneic OSCC mouse models enable mouse tumors with a mouse cell microenvironment to be established, making it possible to investigate the processes of the tumor formation and its interaction with the TME cells. However, to assess the effectiveness of immunotherapeutic treatments, models using both human tumors and immune cells are required as the mouse immune system does not always correspond to that found in humans [135]. The demand for such models is seen in the high failure rate of therapeutics between preclinical mice trials and subsequent human trials [136].

Human xenograft models represent an alternative and are the transplantation of human cells into an immunocompromised host. Athymic nude mice or severe-combined-immunodeficiency (SCID) mice are often used for human xenografts and can be used to test the effectiveness of cytotoxic drugs [137,138]. However, the lack of immunity makes this model unsuitable for the direct investigation of immunotherapy.

Athymic nude mice (classic) have severe T cell dysfunction but still have B cells, as well as neutrophils, dendritic cells (DCs), NK cells, and other components of the innate immune system [139]. SCID mice are deficient in DNA-dependent protein kinases and thus affect T- and B-cell development [140]. SCID mice have a higher human tumor engraftment efficiency than nude mice [141]. In addition, SCID mice are also the first strain to be injected with PBMCs or to be transplanted with human hematopoietic stem cells (HSCs) for the development of humanized mouse models [142,143].

Regarding the host of the humanized model, the premise of establishing a humanized mouse with a human immune system is the construction of immunodeficient mice, followed by continuous optimization and development, because only by destroying the recipient mouse’s immune system first can the human tissue or cells transplanted into its body rebuild a functional human immune system. From the earliest nude mice (nude) to the later SCID mice and Rag^−/−^ mice to NOD-SCID mice, these early immunodeficient mice, either due to the presence of innate immunity resulting in a low success rate of human cell engraftment or due to a high sensitivity to irradiation or due to a shorter life span, all have limited the application of humanized mice in research in varying degrees [143,144,145,146,147].

The development of immunodeficient mice ushered in a milestone breakthrough after 2000 AD. It has been found that interfering with the deletion of the interleukin 2 (IL-2) receptor gamma chain (IL-2Rγc) not only causes severe defects in mouse T and B cells but also interferes with the development of NK cells, which can further improve the transplantation effect of human cells into mice. On this basis, a variety of more advanced severe immunodeficiency mice have been bred by researchers around the world and become the most widely used humanized mouse model hosts for immunotherapies, including BALB/c-Rag2^−/−^γc^−/−^(BRG), NOD/shi-SCID γc^−/−^ (NOG), NOD/SCID-γc^−/−^ (NSG), etc. [148,149,150,151].

Regarding the establishment of a humanized model, according to the reconstruction method of the human immune system, two types of humanized mouse models are commonly used:

(1) The Hu-PBMC mouse model

Peripheral blood mononuclear cells (PBMC) are cells in the peripheral blood with a single nucleus composed of lymphocytes (T or B cells), monocytes, phagocytes, dendritic cells, and a small number of other cell types. It is an important cellular component of the body’s immune response function.

The Hu-PBMC model, also known as the Hu-PBL (peripheral blood lymphocyte, PBL) model, is a simple and economic humanized mouse model of the immune system. Constructed by an intraperitoneal (IP) or an intravenous (IV) injection of mature lymphocytes (derived from PBMC) into immunodeficient host mice, this model is often used to study the activation of human effector T cells and to evaluate immunosuppressive drugs.

The preparation period for the Hu-PBMC model is short. After PBMC transplantation, human CD3^+^ T cells can be detected as soon as one week; in about 2 weeks, the immune cells will be rapidly reconstituted; in about 4 weeks, about 50% of human CD45^+^ cells can be detected in the peripheral blood of mice, about 90% of them are CD3^+^ T cells, and the ratio of CD4^+^:CD8^+^ T cells is about 1:1. However, the Hu-PBMC model develops lethal graft-versus-host disease (GvHD), the extent of which is directly related to the engraftment of human T cells, as assessed by weight loss in mice. GvHD generally occurs 2–3 weeks after transplantation, so the experimental observation window is short [152].

(2) The Hu-HSC mouse model

Another approach involves the injection of human CD34^+^ HSCs into immunodeficient host mice, which requires first sublethal irradiation of the host mice to deplete mouse HSCs and to facilitate engraftment of human HSCs. Such models, also called hu-CD34^+^, or hu-SRC (SCID-repopulating cell) models, have been widely used to study human hematopoietic development, cell-mediated immune responses, and viral infectious diseases such as HIV and EBV.

Hematopoietic stem cells (HSCs) are a type of cell in human hematopoietic tissues that can self-renew and differentiate into various blood cells. CD34 antigen is widely recognized as a representative surface marker of hematopoietic stem/progenitor cells. It is a highly glycosylated type I transmembrane glycoprotein that regulates cell adhesion and promotes cell adhesion to the bone marrow matrix.

A variety of hematopoietic stem cells can be generated by intravenous (IV) or intrafemoral (IF) injection of human CD34^+^ HSCs from human umbilical cord blood, bone marrow, G-CSF-activated peripheral blood, or fetal liver into adult immunodeficient mice, but T cells are produced in small quantities and are not functional. Alternatively, by intravenous injection (intracardiac or intrahepatic), the transplantation of human CD34^+^ HSCs into neonatal recipient mice (less than 4 weeks old) results in good human cell engraftment and production of NK cells, macrophages, B cells, T cells, and DCs. Embryonic liver and umbilical cord blood are the most used sources of human CD34^+^ HSCs, which are more likely to colonize immunodeficient mice than adult HSCs [152].

For tumor research, the establishment of a humanized xenograft mouse model that allows interaction between the human tumor-immune system and the use of novel mouse models to explore the treatment of immune checkpoint inhibitors are currently emerging development directions.

### 5.1. Humanized Xenograft Mice

Humanized mouse models with co-transplantation of the human immune system and human tumors (xenografts) are effective tools for developing new strategies for tumor immunotherapy. If classified by the source of the graft, xenograft models include CDX, which is derived from tumor cells cultured in vitro, and PDX, which is derived from fresh tumor tissue from patients. If classified by the location of inoculation, xenograft models include both ectopic and orthotopic. In ectopic models, tumor cells are injected subcutaneously into the flank or back of the mouse. In contrast, in orthotopic models, tumor cells are often grafted onto the tongue of the mouse [70,153]. Implanting human tumor cells, CDX, or tumor tissue, PDX, into humanized mice based on the human immune system can better simulate the human tumor microenvironment. It is used to study the growth of tumors in the human immune environment and it can evaluate the efficacy and related mechanisms of anti-tumor therapy, especially immunotherapy [154]. Tumor cell or tissue inoculation is performed on the Hu-PBMC model or Hu-HSC model to establish tumor-bearing humanized mouse models, which can be used to evaluate the therapeutic effects of immune checkpoint inhibitors or dual checkpoint inhibitors:

(a) Assessing the effects of immune checkpoint inhibitor therapy

A growing number of studies have shown that humanized mouse models of the immune system have unique advantages in the study of immune checkpoint inhibitors; for example, inoculating KHOS cells in Hu-PBMC mice to establish a humanized osteosarcoma mouse model of the immune system. Moreover, using this model to prove that PD-1 inhibitors can effectively inhibit the lung metastasis of osteosarcoma. The anti-PD-1 antibody reduced tumor size by more than 50% in Hu-PBMC mice inoculated with human lymphoma SCC-3 cells or glioblastoma U87 cell models. The humanized mouse tumor-bearing model of the immune system is a good platform for evaluating immune checkpoint inhibitors [155].

(b) Assess the effect of combination therapy, e.g., dual checkpoint inhibitors

Hu-PBMC mice were inoculated with human colorectal HT-29 cells and gastric cancer tissue and treated with urelumab (anti-hCD137) or/and nivolumab (anti-PD-1). It was found that the combination therapy or monoclonal antibody alone can slow down tumor growth, the combination therapy did not significantly improve the efficacy, and the tumor grafts increased IFNγ-secreting human T cells and decreased human Treg cells. In the Hu-HSC model of EBV-associated lymphoma, the combination of PD-1 and CTLA-4 inhibitors can effectively inhibit the growth of EBV-induced diffuse large B-cell lymphoma and the anti-tumor effect is better than that of monotherapy [156,157].

Morton et al. successfully generated the TME in a humanized PDX mouse model of HNSCC. In xenochimeric mice (XactMice), tumors contained human cells derived from humanized bone marrow and a dynamic microenvironment containing human B and T cell populations, cytokine expression, and lymphangiogenesis. These mice accurately represented the in vivo growth of the original tumor [67]. This HNSCC humanized mouse model was further optimized by Morton et al. [68]. Through the infusion of mesenchymal stem cells (MSCs) and human hematopoietic stem and progenitor cells (HSPCs), the percentage of human immune cells present in the bone marrow of established mice could be doubled compared with mice only implanted with HSPCs. In addition, a 9–38-fold increase in mature peripheral human immune cells was seen for the dual infusion, and these mice had more MSCs, cytotoxic T cells, and regulatory T cells. Therefore, the dual infusion of MSCs and HSPCs produced a higher degree of humanization, which improved the accuracy of the model.

The humanized mouse model is an emerging technology and thus still has many limitations. For example, tissue incompatibility may cause immune responses in humanized mice [158]. This suggestion is supported by findings that transplanted human immune cells, predominantly T cells, produce GvHD and attack recipient mice, resulting in death. As a result, only a short period is available to perform experiments [159]. Nonetheless, the humanized mouse is the closest to replicating human disease, thus ongoing development of this technology is needed.

### 5.2. Humanized Immune Checkpoint Knock-In Mice

Humanized xenografted mice can be prohibitively expensive and laborious to develop. However, an alternative method is the introduction of knock-in (KI) human immune checkpoint genes into mice, especially when it comes to testing drugs that target the immune checkpoints of human immune cells or human tumor cells [160].

An antigenic immune response is regulated by a balance between inhibitory and stimulatory signals through signaling pathways of receptors and ligands called immune checkpoints. Several checkpoint molecules, for example CTLA-4 and PD-1, can affect the functionality of both adaptive and innate immune cells [161]. Tumor cells manipulate these pathways by either suppressing stimulatory immune regulators or overexpressing inhibitory immune checkpoints. Immune checkpoints are an area of interest to immuno-oncology, which seeks to modulate the immune system to identify and destroy cancerous cells [162]. Immune checkpoint blockers (ICBs) are a novel field in drug research and are developed to block ligand–receptor binding between T cells and cancer cells [163].

The differences in the immune systems between humans and mice (60% homology) can cause discrepancies between preclinical and clinical studies. For example, the inhibitory receptors on NK cells that recognize MHC class I molecules differ and do not exhibit cross-reactivity (KIRs in humans and Ly49s in mice). Therefore, studying these receptors in immunocompetent mouse models is challenging; murine Ly49s do not bind to human MHC and vice versa [163,164]. Therefore, early research into NK/ILC checkpoint inhibitors depended on immunodeficient mice transplanted with human tumor cells and only one type of human immune cell. Thus, better preclinical models that mimic human cancer development and our immune system will provide a more realistic TME to measure the success of ICBs. To connect the laboratory with clinical translation, several ready-made mouse models in which human immune checkpoint orthologs and immune regulatory genes are knocked in to replace all or part of endogenous mouse genes.

Murine cells can be edited to express a human form of a murine protein while still being regulated by murine elements. Therefore, these edited mice have a single gene alternation to their immune system, retaining the overall intracellular signaling [165,166]. These mice are called human immune checkpoint knock-in (KI) mice. These mice do not possess human immune cells, so they are not viewed as human immune system (HIS) mice. Nevertheless, these mice can be used to assess the anti-tumor and binding and efficacy of antibodies against discreet human checkpoint molecules. The human immune checkpoint KI mice can further be used to establish HIS mice, resulting in humanized immune checkpoint KI mice. These humanized knock-in immune checkpoint mouse models are ideal for producing tumor mouse models for drug testing and basic research. Commercially available mice can be purchased that express human CTLA-4, human PD-1 and CTLA-4, or both human PD-L1 and PD-1 without murine PD-L1 or PD-1 [162]. These models can help assess the direct and secondary effects of targeted antibody therapies on tumor growth in vivo, reduce off-target toxicities, and optimize antibody binding kinetics to reduce the incidence of severe immunotherapy-related adverse events.

The humanized immune-checkpoint mice model has some other advantages as follows:

Precise humanized gene insertion: mitigates against any possible interference with the humanized gene expression. The target protein confirmation, in particular the binding region, remains unaffected.

Highly efficient models: such models undergo comprehensive QC and validation using clinical-grade checkpoint blockers to ensure adequate expression and correct signal transduction of human immune checkpoint genes.

Fully functional and potent immune system: the murine immune system is not compromised; instead, specific immune cells will express the human version of checkpoints of interest.

## 6. Conclusions and Future Perspectives

Immunotherapies are changing the treatment landscape for oral cancer. Compared with current treatments, immunotherapies improve survival and reduce toxicity. ICIs have now been advanced from second-line to first-line therapy for recurrent or metastatic (R/M) oral cancer. The mouse models used in related studies have various advantages and disadvantages. Syngeneic, GEM, and humanized PDX models all have several limitations. When it comes to recapitulating the heterogeneous anti-tumor immune responses observed in immuno-oncology clinical trials, the limitations mentioned above pose challenges for any single model type. However, further development of more specialized preclinical models will enable us to define their selection to accurately replicate the human malignancies and to better assess responsiveness to immunotherapy [74]. Accurately determining the initial experimental question is crucial when choosing an appropriate model.

Although current research primarily relies on syngeneic tumor models, increasingly complex GEMMs models can accurately replicate the autochthonous tumor microenvironment, enhancing the predictive value of tumor immunotherapy in the preclinical setting. Additionally, advancements in humanized PDX models can potentially develop a “co-clinical” approach to engraft patient tumors into preclinical models of humanized immune reconstitution to guide treatment decisions [74]. While generating humanized PDX models has focused on recreating the patient’s immune system, a crucial advancement will be incorporating microbiome analysis into preclinical models. Given the sensitivity of the microbiome, which can be affected by, among others, the manufacturer and the study housing environment, it is important to include these variables when measuring the immunotherapeutic response [167].

Another critical area of preclinical tumor models that needs further development is the accurate modeling of immune-related adverse events (irAEs), as available preclinical models do not replicate the severity, kinetics, and nature of the toxicities observed in ICB-treated patients. Because immuno-oncology drugs can present unpredictable patterns of irAEs in patients, identifying the early biomarkers of irAEs and approaches for reversing lethal toxicities are crucial for the safety of immuno-oncology combination treatments in the clinic [168]. Such improvements include using models that have reduced kinetics of tumor regression (to facilitate the development of irAEs), incorporating novel ways to determine toxicity, as well as assessing these treatments in models more sensitive to autoimmunity (Treg-depleted model systems [169]). Because irAEs can have a multitude of manifestations in various tumor histologies, incorporating models that correctly present the irAEs seen in the clinic is of the utmost importance [170].

## Figures and Tables

**Figure 1 ijms-23-09195-f001:**
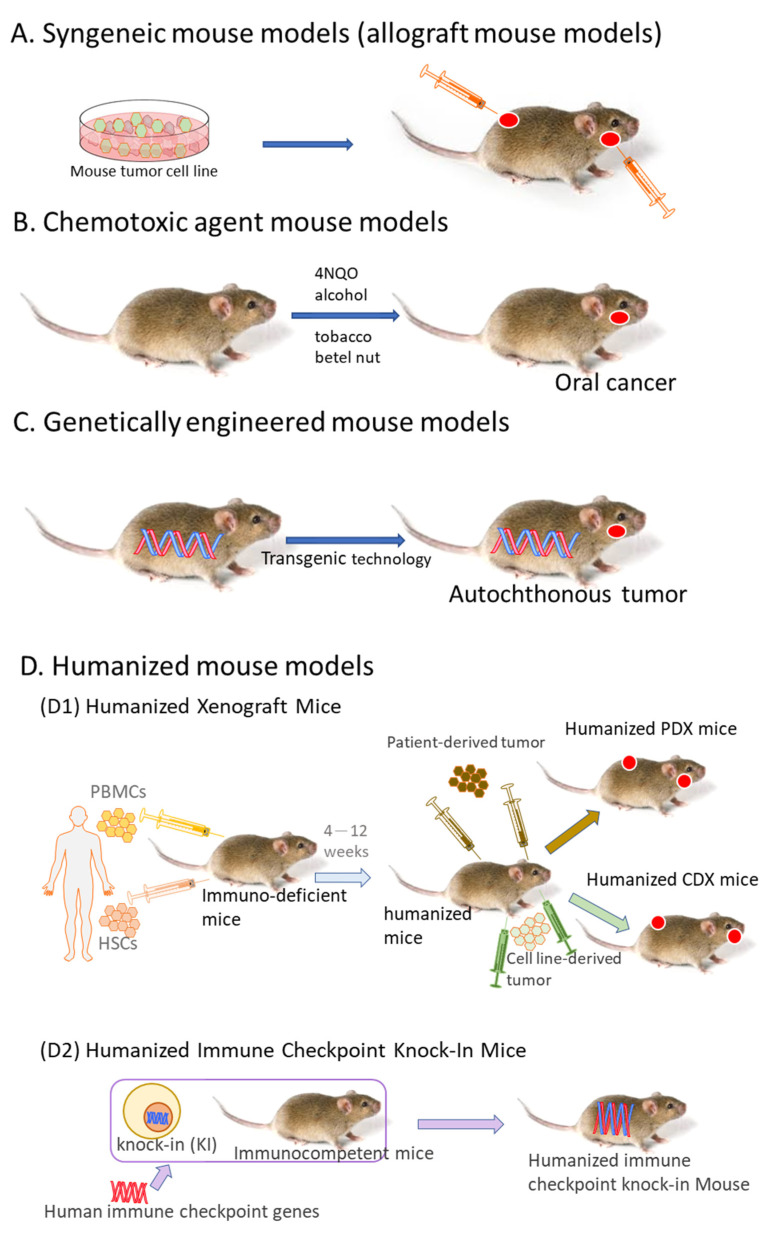
Murine models for oral tumor immunotherapy research. (**A**) Syngeneic tumor models use murine tumor cell lines grown and expanded in vitro that are then injected (subcutaneously or orthotopically) into immunocompetent hosts. (**B**) Chemotoxic agent mouse models: Chemicals (carcinogens) are administered to induce OSCC growth. In mice treated with carcinogens, the tumor also forms spontaneously de novo. (**C**) genetically engineered mouse models use autochthonous tumor cell growth driven by the tissue-specific deletion of tumor suppressors or tissue-specific expression of oncogenes. (**D**) The humanized mouse can be further divided into two categories: (**D1**) humanized xenograft mice (peripheral blood mononuclear cells (PBMCs) or hematopoietic stem cells (HSCs) were implanted into immunodeficient mice. After 4–12 weeks, patient-derived tumors or cell line-derived tumors were implanted into mice orthotopically or subcutaneously to form humanized patient-derived xenograft (PDX) mice or humanized cell-derived xenograft (CDX) mice, respectively); (**D2**) using the expertise in the genetically engineered mouse model (GEMM) generation by CRISPR/Cas9, human immune checkpoint knock-in (KI) mice were generated that can be used to test specific molecular interactions between tumor cells and immune cells as well as within the immune system. The human immune checkpoint KI mice can further be used to establish human immune system (HIS) mice, resulting in humanized immune checkpoint KI mice. 4-NQO—4-nitroquinoline 1-oxide.

**Table 1 ijms-23-09195-t001:** Phase III trials evaluating alone or combination of immune checkpoint inhibitors in HNSCC.

Trial (Code)	Drugs (Brand Name) Interventions	Patients Number	Immune Checkpoint Tested	
KEYNOTE-048 NCT02358031	Pembrolizumab ± chemotherapy vs. chemotherapy + cetuximab	882 [30]	Pembrolizumab: PD-1	First line treatment for R/M HNSCC
KEYNOTE-040 NCT02252042	Pembrolizumab vs. methotrexate, docetaxel or cetuximab	495 [31]	Pembrolizumab: PD-1	Second line treatment for R/M HNSCC
CheckMate 141 NCT02105636	Nivolumab vs. chemotherapy	361 [32]	Nivolumab: PD-1	Second line treatment for R/M HNSCC
EAGLE NCT02369874	Durvalumab ± tremelimumab vs. chemotherapy	736 [33]	Durvalumab: PD-L1 Tremelimumab: CTLA-4	Second line treatment for R/M HNSCC
NCT01836029	Pembrolizumab	195 [34]	Pembrolizumab: PD-1	R/M HNSCC
CheckMate 651 NCT02741570	Ipilumumab and nivolumab vs. cetuximab with platinum and fluorouracil	947 [35]	Ipilimumab: CTLA-4 Nivolumab: PD-1	First line treatment for HNSCC
KESTREL NCT02551159	Durvalumab and tremelimumab vs. Durvalumab monotherapy	823	Durvalumab: PD-L1 Tremelimumab: CTLA-4	R/M HNSCC
NCT03673735	Durvalumab before CRT and every four weeks for six months after CRT. Control: placebo before CRT and six months every four weeks after CRT Radiotherapy	650	Durvalumab: PD-L1	R/M HNSCC

R/M—recurrent or metastatic; PD-1—programmed cell death protein 1; PD-L1—programmed cell death 1 ligand 1; CTLA-4—the cytotoxic T-lymphocyte-associated antigen 4; CRT—chemoradiotherapy.

**Table 2 ijms-23-09195-t002:** The pros and cons of murine models for immunotherapy research in oral tumors.

Model	Pros	Cons
**Syngeneic**	Low in cost. Easy to set up. Rapid tumor development. Fully functional mouse immune system.	Tumors do not develop a natural microenvironment. Tumor heterogeneity is low. There are major differences between the immune systems of mice and humans. May cause vaccination effects.
**Chemotoxic agent**	Development of precancerous lesions. Host-tumor cell interactions are conserved. Tumors form a natural microenvironment. Fully functional mouse immune system. Tumor heterogeneity is high. Easy to use. Sporadic cancer development. It can be used in combination with other tumor induction methods.	High in cost. Difficult to set up. Time-consuming. Laborious. Tumor formation is not initiated by chronic inflammation. Chemical hazards. Metastasis and bone invasion are rare. Difficult to monitor tumors. Variability in tumor progression time. Due to high heterogeneity, larger sample sizes were required for data interpretation.
**Genetically engineered**	Gene expression can be manipulated. Useful for the study of genetic alterations. Encompasses natural tumor microenvironments. The tumor forms a natural microenvironment. Fully functional mouse immune system. The genetic and histopathological aspects of all stages of cancer can be recapitulated. Heterogeneity of the tumor is higher than in syngeneic models, depending on the production method.	High in cost. Overexpression of transgene. Tumors unrelated to the oral cavity may develop. Labor intensive. Limited accessibility. Difficult to monitor a tumor. Low immunogenicity. Challenging for breeding and gene manipulation. Homogeneous in the genomic aspect.
Humanized	Highly reflect promoter methylation in tumors and reproduced tumor heterogeneity. Heterogeneity of the tumor is high.	High in cost. Difficult to set up. A partial TME is transplanted from the patient and is dependent on the transplantation site and donor immune cells.
Hu-PBMC	Simple to construct, the T cell transplantation efficiency is high and it is stable.	Limited study time due to short survival as well as the occurrence of GvHD.
Hu-HSC	Development of multilineage hematopoietic cells, including T cells, B cells, NK cells, and myeloid cells.	T cells are educated by the mouse thymus; T cells are few and non-functional, not HLA-restricted.
Humanized Immune Checkpoint Knock-In Mice	Human gene knock-in mice strains allow for a robust expansion of human immune cells in the mouse TME. Precise humanized gene loci. Fully functional and potent immune system.	Often the transplant is subcutaneous, resulting in the surrounding environment lacking the chronic inflammatory milieu and organ-specific factors of the tumor.

peripheral blood mononuclear cells (PBMCs); hematopoietic stem cells (HSCs); tumor microenvironment (TME); graft versus host disease (GvHD).

**Table 3 ijms-23-09195-t003:** Mouse models used for immunotherapy in oral tumors.

Model	Animal Background	Inducer	Dosage/Treatment Period	Tumor Harvest/Formation/End Point/Conclusions/Development Period (Weeks)	References
**Syngeneic**	C3H	SQ-1979 (Subcutaneous; Orthotopic); SCC7 (Subcutaneous)	5 × 10^6^~1 × 10^7^ cells; 1 × 10^5^~1 × 10^6^ cells	3 weeks	[36,37,38,39]
	C57BL/6 (Orthotopic)(Subcutaneous)	MTCQ1, MTCQ2, MOC-L1, MOC-L2, MOC-L3, MOC-L4, NHRI-HN1, NHRI-HN2, MOC1, MOC2-luc, MOC2 mKate2	1.5 × 10^5^~5 × 10^6^ cells	2~5 weeks	[40,41,42,43,44,45]
	BALB/cAnN.Cg-Foxn1^nu^/CrlNarl (Subcutaneous)	NHRI-HN1, NHRI-HN2	1 × 10^6^ cells	6 weeks	[45]
**Chemotoxic agent**	C3H (Orthotopic); C57BL/6 (Orthotopic); BALB/c CF-1 (Orthotopic); CBA (Orthotopic)	4-NQO	50~100 μg/mL/4~20 weeks	4~70 weeks	[46,47,48,49,50,51,52,53,54,55]
	C57BL/6J (Orthotopic) (male wild-type) 5-Lox knockout	4NQO + Alcohol	100 μg/mL + 8%/8 weeks + 16 weeks		[56]
	C57BL/6JNarl (Orthotopic)	4NQO + Arecoline	200 μg/mL + 500 μg/mL/8 weeks		[57]
	B6C3F1 (Orthotopic)	Tobacco-related (1) DB(a,l)P (2) B(a)P (3)N’-nitrosonornicotine (NNN)	24 nmol 100 ppm/2 years 8.46 μmoL/2 times/week, 4 weeks		[58,59,60]
**Genetically engineered**	L2D1^+^ /*p*53^+/−^ and L2D1^+^ /*p*53^−/−^ (Orthotopic)			Formation of invasive oral–esophageal SCC at 6 months.	[61]
	LSL-*Kras*^G12D^ (Orthotopic)	K5 or K14-CrePR1 CrePR1		Oncogenic K-*ras* ^G12D^ overexpression induced in oral epithelium of mice by16–24 weeks administration of RU486. Formation of squamous papilloma in the oral cavity.	[62]
	*Tgfbr1/Pten* 2cKO mice (Orthotopic)			Tgfbr1/Pten 2cKO mice induced with 10-week administration of tamoxifen (tam). Formation of cancer and precancerous lesions in the oral epithelium.	[63]
	*p53^R172H^*; K5-CrePR1 and *p53^flflox/flflox^*; K5-CrePR1 (Orthotopic)			Formation of OSCC at 15–16 months: p53flflox/flflox; K5-CrePR1 (25%) and p53 R172H; K5-CrePR1 (16%) mice	[64]
	LSL-*Kras* (Orthotopic); iHPV-Luc (Orthotopic); K14-CreER^tam^ mice (KHR mice) and LSL-*Kras* (Orthotopic); K14-CreER^tam^ mice (KR mice) (Orthotopic)			Formation of oral tumors in KR and KHR mice using tamoxifen. Bioluminescence signal of KHR mice 74.8 times higher than control mice.	[65]
	HPV16 E7iresE6 (Orthotopic); PIK3CA ^E545K^ (Orthotopic); KRT14-CreER^tam^ mice (Orthotopic)			After the administration of tamoxifen for 6–8 weeks, oropharyngeal tumors developed with about 40% penetrance (1–2 tumors/tongue).	[66]
**Humanized**	XactMice (Orthotopic)	SHPCs, MSCs			[67,68]
	(1) *BALB/c-Rag2^null^* (Orthotopic); *IL2rg^null^ (BRG)* (Orthotopic) (2) *NOD.Cg-Rag1^tm1Mom^ IL2rg^tm1Wjl^ Ins2Akita (NRG-Akita)* (Orthotopic) (3) *NOD.Cg-Prkdc^scid^ Il2rg^tm1Wjl^ (NSG)* (Orthotopic)	CD34^+^ umbilical cord blood cells	(1) 1 × 10^5^ cells (2) 5 × 10^4^ cells (3) 2 × 10^5^ cells	(1) Total of 300–500 human islets transplanted subrenal capsule 8–26 weeks post CD34 HSC engraftment into normoglycemic BRG mice. (2) Total of 4000 human islets transplanted subrenal capsule into diabetic NRG-Akita mice. (3) Total of 3000–4000 human islets transplanted subrenal capsule into diabetic NSG mice, mice treated with streptozotocin to induce diabetes (timing not reported in relation to HSC engraftment or islet transplantation).	[69]

K5 or K14-CrePR1—keratin 5 (K5) or keratin 14 (K14) tissue-specific promoter used in the overexpression of the oncogene K-*ras*^G12D^ targeted to the oral epithelium of mice. CrePR1—Cre recombinase fused to a human progesterone receptor mutant.

## Data Availability

Not applicable.

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
