# Peer review of "Mouse Models for Immune Checkpoint Blockade Therapeutic Research in Oral Cancer"

_ijms, 2022, doi:10.3390/ijms23169195_

Round 1

Reviewer 1 Report

This is a nice and original literature review regarding the mouse models for immune checkpoint inhibitors for oral cancer.                                                                                                               The survey of the literature is satisfactory and references are adequate.

Author Response

Thank you very much for your comments, please also refer to the attachment.

Reviewer 2 Report

In this manuscript, the authors reviewed the oral squamous cell carcinoma (OSCC) mouse models for immune checkpoint blockade drugs, such as immune checkpoint inhibitors (ICIs) and related treatments. First, the authors raised the point that good in vivo models, such as mouse models, are relatively low cost and can form tumors easily. Second, the authors did a comprehensive review of the existing mouse models: syngeneic tumor mouse models, chemical carcinogen-induced models, genetically engineered mouse models, and humanized mouse models. By summarizing the advantages and disadvantages of these tumor models from existing literature, this review could provide help when choosing a mouse model for related OSCC research. 

One minor thing to change is in the abstract, the order of the mouse models should be in line as in the main manuscript, not in the revered order.

Author Response

Thank you very much for your comments; please also refer to the attachment.

Reviewer 3 Report

In their review "Mouse Models for Immune Checkpoint Blockade Therapeutic Research in Oral Cancer" the authors  present evidence and considerations for choosing a suitable model establishment method to investigate the early diagnosis, clinical treatment, and related pathogenesis of OSCC.

The review is very interesting and provides useful information about a current need in medicine. 

I would sugguste to make the folliowng corrections/additions to improve the quality of this review.

- Figure 1: needs some stylish workup (alignement of text, placement of text in graphic, colour selection, font size ....

- Table 1: here it would be helpful to make a graphical presentation of the results of the different studies - how do they compare - or how are they comparable - statistical analysis of the studies: what were the outcomes?

After these corrections, I suggest for publication

Author Response

(The authors gave the same response as above.)
